# A Dual Hollow Core Antiresonant Optical Fiber Coupler Based on a Highly Birefringent Structure-Numerical Design and Analysis

**Hanna Izabela Stawska *** and **Maciej Andrzej Popenda**

Department of Telecommunications and Teleinformatics, Wroclaw University of Science and Technology, 50-370 Wroclaw, Poland; maciej.popenda@pwr.edu.pl
* Correspondence: hanna.stawska@pwr.edu.pl; Tel.: +48-71-340-76-42

**Abstract:** With the growing interest in hollow-core antiresonant fibers (HC-ARF), attributed to the development of their fabrication technology, the appearance of more sophisticated structures is understandable. One of the recently advancing concepts is that of dual hollow-core antiresonant fibers, which have the potential to be used as optical fiber couplers. In the following paper, a design of a dual hollow-core antiresonant fiber (DHC-ARF) acting as a polarization fiber coupler is presented. The structure is based on a highly birefringent hollow-core fiber design, which is proven to be a promising solution for the purpose of propagation of polarized signals. The design of an optimized DHC-ARF with asymmetrical cores is proposed, together with analysis of its essential coupling parameters, such as the extinction ratio, coupling length ratio, and coupling strength. The latter two for the $x$- and $y$-polarized signals were ~2 and 1, respectively, while the optical losses were below 0.3 dB/cm in the 1500–1700 nm transmission band.

**Keywords:** hollow-core antiresonant fibers; optical fiber couplers; hollow-core optical fiber couplers; optical fiber design; polarization splitters

## 1. Introduction

Over twenty years have passed since Cregan et al. presented the first hollow-core optical fiber (HCF) with a microstructured cladding [1]. Remarkable optical features of HCFs [2–4] have resulted in their extensive experimental use, for example in telecommunications [5–7], gas and liquid spectroscopy [8–13], supercontinuum generation [14,15], high power optical beam delivery [16–18], transmission in the spectral regions unavailable for conventional fibers [19–22], biomedical applications [23,24], and many others. Although the overall scientific reach of both types of HCFs—namely hollow-core photonic bandgap and hollow-core antiresonant optical fibers (HC-PBFs and HC-ARFs, respectively)—has vastly exceeded that of conventional, step index fibers, their full potential is still to be discovered. Indeed, one of the reasons for this is the lack of HCF-integrated optical devices, such as optical fiber couplers (OFCs) based on a structure with two closely spaced fiber cores. Initial research in this area revolved around the solid-core photonic crystal fibers (SC-PCFs). In 2000, Mangan et al. fabricated and measured the transmission spectrum of the first dual solid-core photonic crystal fiber (DSC-PCF) [25]. One important observation of this paper was that the coupling spectrum (i.e., the total signal intensity in a given core of the DSC-PCF versus the signal wavelength) of the DSC-PCF was a function of $\Lambda$, the pitch of the fiber's microstructured cladding. This result indicated the possibility of a straightforward modification of coupling characteristics of DSC-PCFs simply by modifying the parameters of the microstructure itself. In the following years the topic of coupling in DSC-PCFs has been studied in both numerical and experimental manners [26–28]. DSC-PCF has also been made

with a lead-silicate glass and used for the purpose of supercontinuum generation [29]. However, the solid core of all the above structures was their ultimate limitation, resulting in susceptibility to optical nonlinearities, transmission bands dependent on the core's transparency, and so on. Until now, amongst various theoretical papers [30–32], there has been only a single experimental report of a dual hollow-core photonic bandgap fiber (DHC-PBF) [33]. However, since the coupling between both its cores was strongly dependent on the external pressure applied by bending the fiber, its practical use was quite cumbersome.

Previously mentioned HC-ARFs, on the other hand, seem to be a more potent platform for the design and fabrication of OFCs. These fibers are known for their multiple, wide transmission windows and remarkably high core-mode confinement, which reduces their dispersion to only a few ps/nm × km. All of the above, combined with a great modification and design flexibility of HC-ARFs, have resulted in researchers presenting numerous different structures based on two most known HC-ARF cladding types—the single-ring [34,35] and Kagomé [36,37] types. This flexibility can also be considered one of the factors which have encouraged researchers to pursue the idea of a dual hollow-core antiresonant fiber (DHC-ARF). In 2009, Argyros et al. presented a polymer DHC-ARF with two transmission bands in the UV-NIR (Ultraviolet-Near infrared) [38]. Coupling between the core modes was strongly suppressed within those bands due to negligible overlap of the mode fields of both cores. The latter is a consequence of the previously mentioned high core-mode confinement observed for HC-ARFs. As a result, efficient coupling between both cores of the presented polymer DHC-ARF was possible only at the edges of its transmission bands, where the signal starts to leak through the Kagomé structure. However, coupling through the structure also caused the high-order and cladding modes to appear, making the proposed DHC-ARF impractical in use as an OFC. A few years later, a silica Kagomé DHC-ARF was presented [39]. Results from this work have confirmed that key factors for the efficient coupling between the cores of DHC-ARF are core separation distance and core uniformity, as stated in [38]. The researchers have also noticed that the output beam pattern was strongly dependent on the input beam's polarization, but no analysis of polarization-dependent parameters, such as polarization coupling length or polarization coupling ratio, was conducted,. Nevertheless, both those works have proven the feasibility of DHC-ARFs and encouraged further work on the topic, focused on the single-ring structures. A theoretical proposal of a single-ring DHC-ARF was presented in 2016 by Liu et al. [40]. By removing two adjacent cladding ring capillaries (one for each core), the distance between the cores and the localization of the core modes were both decreased, allowing the mode fields to leak into the so-called "coupling-channel". The width (or diameter) of this channel was found to be even more important for the coupling efficiency of DHC-ARFs than the distance between the cores. Additionally, this strongly influences the coupling of signals with different polarizations, suggesting the possibility of creating polarization OFCs. A single-ring cladding DHC-ARF was presented one year later [41], confirming many of the theoretical findings from the previous works. However, since the coupling between both cores was weak (coupling length was 40 cm, independent of the signal's polarization), additional longitudinal tension was required in order to increase the coupling strength. Still, the authorshave presented the potential use of this particular DHC-ARF in fiber lasers [41] and as an integrated fiber-optic interferometer [42,43].

In this paper we present a concept of a DHC-ARF coupler suitable for the purpose of splitting orthogonal polarizations of the input signal. Using the nested capillary, high birefringence HC-ARF concept [44], we designed a DHC-ARF structure capable of efficient coupling of the *x*-polarized component of the input signal between the cores. This fiber's structure combines all the previously referenced knowledge on the topic of DHC-ARFs, so that low loss, high birefringence, and an optimal core proximity could be achieved within a single structure. Although the idea of a polarization-splitting DHC-ARF was presented recently [45], it lacked the computational analysis of propagation of signals with different polarizations. In this paper, we show the importance and necessity of such analysis for the optimization of the DHC-ARF performance in terms of coupling. Starting from a completely symmetrical structure, we slowly introduce geometrical changes affecting both the birefringence and

the proximity of the two cores, finally receiving an optimized model of polarization-splitting OFC based on a DHC-ARF. We have also investigated the effect of the symmetry of both cores on the coupling of signals with different polarizations. In our opinion, the presented approach may not only help in the process of potential future fabrication of such fibers, but it also provides some additional insight into the matter of light guidance in HC-ARFs, which is still waiting to be fully described.

## 2. Basic Formulas and Principle of Operation

By bringing two optical fiber cores into proximity with each other, their mode fields start to overlap and interact with each other, causing the energy transfer to occur between both cores. This effectively makes such structure a single waveguide, supporting up to four so-called supermodes: an odd and even supermode for each of the two orthogonal polarization states. According to the coupled mode theory, modes with the same polarization direction can couple with each other because of the interference between supermodes. The fundamental parameter of coupling properties for dual core fibers is the previously mentioned coupling length $L_c$, which denotes the shortest propagation distance at which maximum power transfer from one core to another will occur [40]:

$$L_c^i = \frac{\pi}{\beta_{even}^i - \beta_{odd}^i},$$

(1)

where $i$ denotes the polarization direction (either $x$ or $y$), while $\beta_{even}$ and $\beta_{odd}$ are propagation constants of the even and odd supermodes, respectively. Since power transfer occurs in a sinusoidal manner from one core to another, it is important for a polarization OFC to have different coupling lengths for every polarization. For a high performance polarization splitter, the ratio of $L_c^x$ and $L_c^y$ should be close to 2 ($L_c^x > L_c^y$) or 0.5 ($L_c^x < L_c^y$), which ensures that the corresponding power transfer maxima for signals with different polarizations will be shifted in phase by $\frac{\pi}{2}$ [46]. According to this, it is convenient to define another parameter called the coupling length ratio *CLR*:

$$CLR = \frac{L_c^x}{L_c^y},$$

(2)

In order to evaluate the performance of the polarization splitter, the extinction ratio parameter *ER* should also be calculated. *ER* can be defined as the normalized power ratio between the *x*- and *y*-polarized signals in the same core:

$$ER = 10 \log_{10}\left(\frac{P_j^x}{P_j^y}\right),$$

(3)

where $P_j^x$ and $P_j^y$ are the *x*- and *y*-polarized signal powers in the *j-th* core ($j$ = A or B).

## 3. Polarization-Splitting DHC-ARF—Analysis of Several Geometrical Parameters

The most common way to control orthogonal polarizations of a signal in an optical fiber is to create a birefringent core. In general, introducing a birefringence into the optical fiber core requires modification of the refractive index profile in either the *x*- or *y*-direction in the plane of the fiber's cross-section. Unfortunately, HC-ARFs were shown to be relatively resistant to the effect of the core's ellipticity (birefringence-wise) [44]. However, in the same paper the researchers found an alternative method, namely a change of the thickness of cladding capillaries walls. Eventually, this allowed them to design a nested-capillary HC-ARF (NARF) with a birefringence of the order of ~$1.5 \times 10^{-4}$ and loss below 1 dB/m in the 1.5 μm wavelength region. Using this NARF as our base structure, we designed three different versions of a DHC-ARF (Figure 1a–c). Similar to [40], two NARF structures were brought together with their contact cladding capillaries removed, in turn forming a mode-coupling channel. To control the channel's width, two smaller capillaries were introduced. The cladding capillaries of

the designed DHC-ARF aligned along the *x*-axis have thickness $w_1$, while the ones aligned along the *y*-axis have thickness $w_2 \neq w_1$. The symmetry of the neighbouring cores is an important feature for the coupling strength in OFCs [38,40]. However, its influence on the coupling of signals with different polarizations in DHC-ARF has not been investigated so far. In order to study it, we began with three structures that differed in the orientation of the introduced capillary thickness modification. As a result, we investigated three versions of DHC-ARF—two with completely symmetrical cores (Figure 1a,b) and one with asymmetrical ones (Figure 1c). The symmetry comes from the fact that the capillaries with altered wall thickness are aligned along the same axis for both cores—either horizontal (structure-*X*, Figure 1a) or vertical (structure-*Y*, Figure 1b). Figure 1c shows an asymmetrical version of the DHC-ARF, structure-*XY*, with the thinned capillaries being oriented in the *x*-direction for one core and in the *y*-direction for the other.

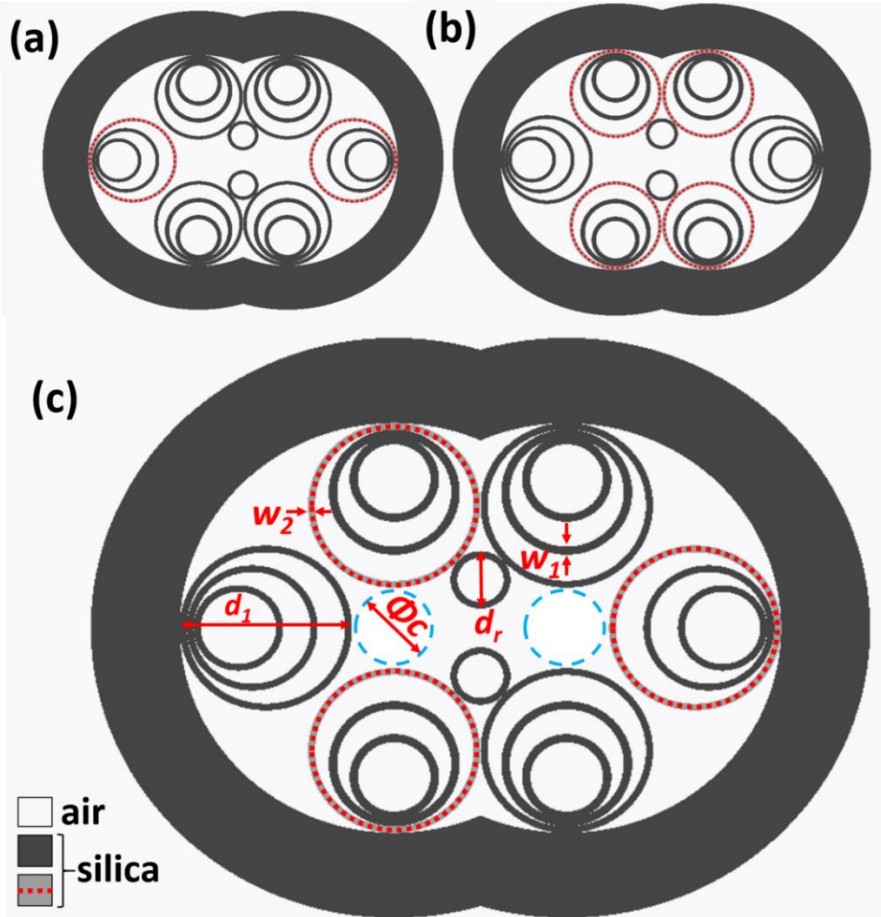

**Figure 1.** Cross-sections of the proposed dual hollow-core antiresonant fibers (DHC-ARFs). (**a**) Structure-*X* and (**b**) structure-*Y*, with capillaries of reduced thickness oriented in their respective directions. Both of those structures can be considered fully symmetrical along their main axes. (**c**) Structure-*XY*, characterized by the increased wall thickness oriented orthogonally to each other for both cores, with one aligned in the *y*-direction and the other one in the *x*-direction. The capillaries with increased thickness are grey colored, with additional red dots inside, while the unchanged ones are black. Additionally, all the relevant dimensions have been marked, with the blue dotted circles indicating the diameter of the DHC-ARF's cores.

In order to minimize the loss, two nested capillaries inside a large one were added and the distance between the inner capillaries was assumed to be $0.65 \times 0.5\Phi_c$ [47], where $0.5\Phi_c = 7$ µm is the radius of the core. The outer diameters of the external (closest to cores) capillaries and small middle capillaries (controlling the width of the coupling channel) are $d_1$ and $d_r$, respectively. As mentioned earlier, cladding capillary thicknesses are denoted as $w_1$ and $w_2$, with $w_2$ being the diameter that was changed during the simulations. Except the latter, the rest of the dimensions are constant and set to $w_1 = 1.172$ µm, $d_1 = 14.3$ µm, and $d_r = 8.8$ µm. In order to investigate the influence of $w_2$ on *CLR*, we introduce the structural parameter $k_1$:

$$k_1 = \frac{w_2}{w_1}, \tag{4}$$

The above parameter is the ratio of the introduced change in the thickness of external capillaries walls. Figure 2 shows the *CLR* and loss in relation to ratio $k_1$ for the three proposed structures. All the simulations presented in this paragraph have been conducted with Lumerical® Mode Solutions commercial software. To properly calculate the loss and mode profiles of the DHC-ARF, the Palik model (included in the Lumerical® material library) was used to determine the dispersion of $SiO_2$. As it can be seen, *CLR* for symmetrical structures never reaches the desired values of 2 or 0.5. For structure-*XY*, for each of the transmission windows, at least one value of $k_1$ can be found for which *CLR* = 2, indicating the possibility of controlling *CLR* by disturbing the symmetry of cores in DHC-ARFs. The relation between loss of the DHC-ARF's fundamental mode (LP01) and the structural parameter $k_1$ is presented in Figure 2b. For the sake of clarity, we show only the highest loss of the fundamental mode LP01. The lowest loss can be observed for structure-*Y* and this result is common for all transmission bands. Additionally, in terms of loss, structure-*Y* is the most sensitive one to the changes of $k_1$ parameter. On the other hand, structure-*X* maintains an almost constant loss level across the whole spectrum of $k_1$ values. This behavior can be explained by the increased leakage loss of structure-*X* through the *y*-oriented capillaries of the structure, which has been observed during the analysis of the modes profiles at $k_1 \approx 0.52$ for both structure-*X* and structure-*Y*. These mode profiles are presented in the inserts of Figure 2b, and one can see that the field amplitudes in the vicinity of the walls of external capillaries are higher for structure-*X*. To ensure this observation, the total power fraction confined in the glass components of the designed DHC-ARF, $\eta$, was calculated according to the following formula [37]:

$$\eta = \iint\limits_{S_{Si}} p_z dS \left( \iint\limits_{S_\infty} p_z dS \right)^{-1}, \tag{5}$$

where $p_z$ is the longitudinal component of the Poynting vector, integrated over the whole structure ($S_\infty$) and its glass components ($S_{Si}$). Values obtained for structure-*X* and structure-*Y* are $9.3 \times 10^{-5}$ and $1.4 \times 10^{-5}$, respectively, directly confirming our initial assumptions about the increased mode leakage in case of structure-*X*. However, since the topic of propagation in HC-ARFs is still an open matter, especially in terms of high-order modes, structure modes, and losses [48–50], this issue still needs to be investigated in the future.

In order to verify the coupling properties of the designed splitter, we have simulated the propagation of the light along the fiber. The simulation was conducted by means of the bidirectional eigenmode expansion (EME) solver, available as a part of the Lumerical® Mode Solutions software. Figure 3 shows the simulation setup. As an excitation fiber, we chose the previously mentioned NARF.

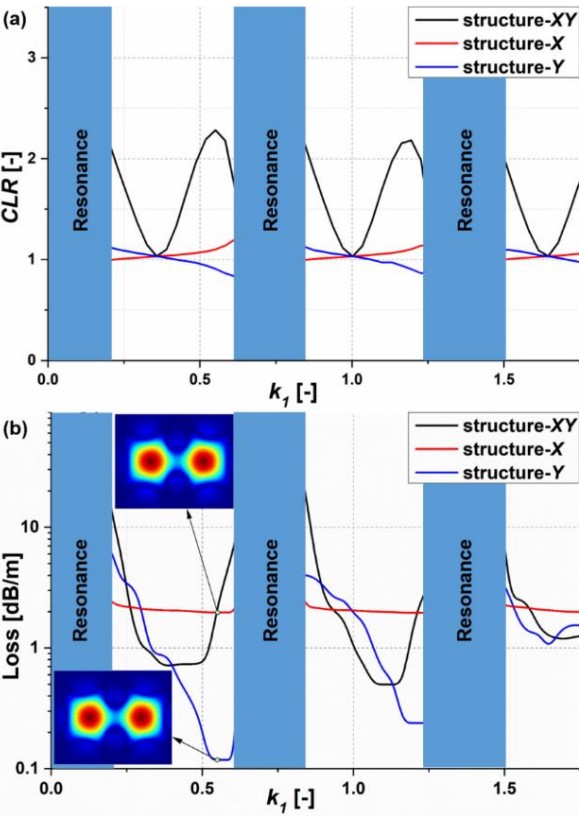

**Figure 2.** (**a**) Coupling length ratio (*CLR*) and (**b**) the highest loss of the fundamental mode LP01 calculated for three presented structures. Inserts show output mode profiles at $k = 0.52$ for structure-*X* and structure-*Y*.

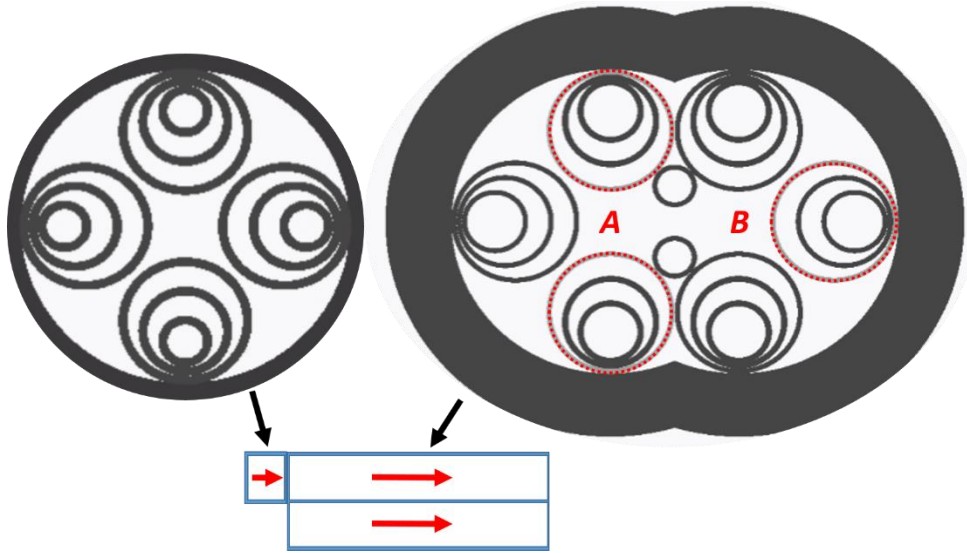

**Figure 3.** Eigenmode expansion (EME) simulation scheme. The fiber to the left is a launch fiber, with a structure previously presented in [44], while the fiber to the right is the proposed polarization-splitting DHC-ARF. The red arrows indicate the direction of the signal propagation. Letters *A* and *B* denote the neighbouring cores of the structure.

A short segment (50 µm) of this fiber was butt-coupled to a segment of the designed DHC-ARF. The latter was excited with a LP01 mode of the NARF. Two cases were simulated—in the first the LP01 mode was *x*-polarized, while in the other it was *y*-polarized. Figure 4 shows the results of these simulations. Although the proposed structure has *CLR* = 2, the total power transfer between the cores, and hence the coupling, is very weak. It is also worth noting that $L_c^x$ = 6.3 mm and $L_c^y$ = 3.15 mm.

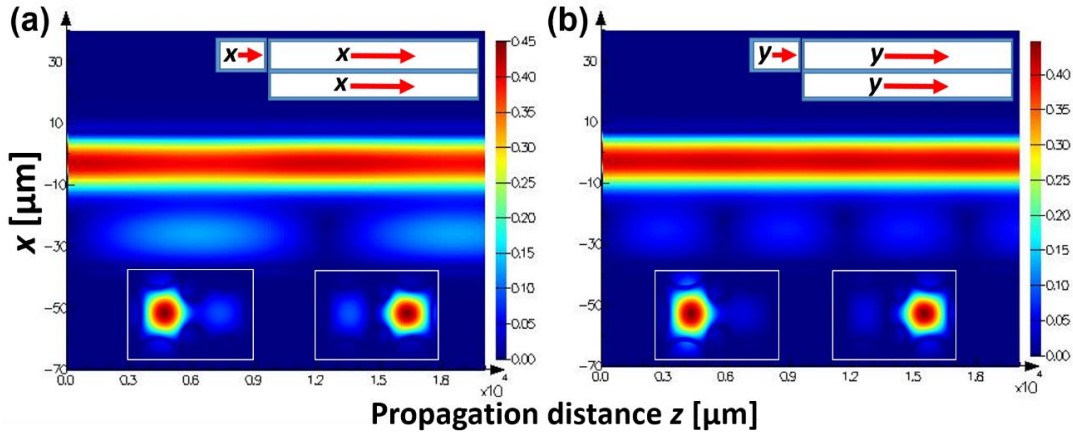

**Figure 4.** Power distribution during propagation along the designed DHC-ARF. The excitation fibre is butt coupled to the core A. **(a)** Light-source polarized in the *x*- and **(b)** *y*-direction. Top inserts of both figures represent the modelled coupling setup, in an identical manner as presented in Figure 3; letters *x* and *y* indicate the polarization direction of the coupled and propagating beam. Bottom inserts show the excited even- (left) and odd- (right) supermodes for both polarizations. Their overlap is slightly stronger in case of the *x*-polarized signal, however, it is still too weak to be considered efficient, even though the *CLR* was optimized to be 2.

According to the coupled mode theory, the fraction of power transferred between the cores is reduced for the dissimilar cores [38]. On the other hand, the output mode power of each core strongly depends on the coupling coefficient *C*, related to the core separation distance $C_s$. For the presented DHC-ARF, the dissimilarity of its cores was shown to be necessary in order to keep the proper *CLR* value. As a result, to overcome the problem of weak coupling, we analyzed the influence of $C_s$ on the coupling strength. In order to conduct this simulation we changed the circular shape of the capillaries aligned along the *y* axis to be elliptical, as shown in Figure 5a. The major and minor axis diameters of the elliptical capillaries are denoted by $d_1$ and $d_2$, respectively. During our simulations, the value of $d_2$ was the only variable. If we assume that the light polarized in the *x*- or *y*-directions is launched into core A, the coupling strength can be defined as follows:

$$SP^{x,y} = \frac{P_B^{x,y}}{P_A^{x,y} + P_B^{x,y}} \tag{6}$$

where $P_A^{x,y}$ and $P_B^{x,y}$ denote the power of the signal in cores A and B, respectively. The power flowing across a particular core for each polarization of the source signal was calculated using the *E* and *H*-field distributions obtained during simulations, based on the following formula:

$$P_{A,B}^{x,y} = \frac{1}{2} \int_{S_{A,B}} \vec{W} * d\vec{S}_{A,B} \tag{7}$$

where $W$ is the Poynting vector $\vec{W} = \vec{E} \times \vec{H}^*$ and $S_{A,B}$ is the surface of core A or B. Calculations were conducted assuming the proper length of the fiber was equal to $L_c^{x,y}$. The simulation results are presented in Figure 5b. Figure 6 shows results for the structure with $d_2 = 10.1$ μm. It can be noticed that mode fields of the LP01 $y$-polarized mode (Figure 6a) have a small overlap, which results in its weak coupling. In the case of the $x$-polarization, the coupling is significantly stronger, and the mode overlap is also bigger. In Figure 6b, the distribution of the power during propagation in core A and B is shown. Two cases are analyzed: in the first, the initial power is launched into core A, and in the other it is launched into core B. Although the fiber has asymmetrical cores, the distribution of power is similar for both cases. Figure 6c shows the calculated loss for LP01 mode. Because only a short segment of the fiber is used as a splitter, the transmission window is relatively wide and equal to 200 nm (from 1500 nm to 1700 nm). Also, the *CLR* is almost constant in this range of wavelengths, as shown in Figure 6d. In order to assess the single mode performance of the proposed fiber, we calculated the higher order mode extinction ratio (HOMER), which can be defined as the ratio of the lowest higher order mode loss to the highest fundamental mode loss [45]:

$$HOMER = \frac{\alpha_{HOM}}{\alpha_{FM}} = 28, \tag{8}$$

where $\alpha_{HOM}$ and $\alpha_{FM}$ are the attenuations (in dB/m) of high-order and fundamental modes, respectively. These calculations were conducted at the wavelength of 1550 nm. In the case of the excitation fiber butt-coupled to core A, we have determined *ER* = 16 dB for core A and *ER* = 17 dB for core B, for a propagation distance equal to 8.3 mm.

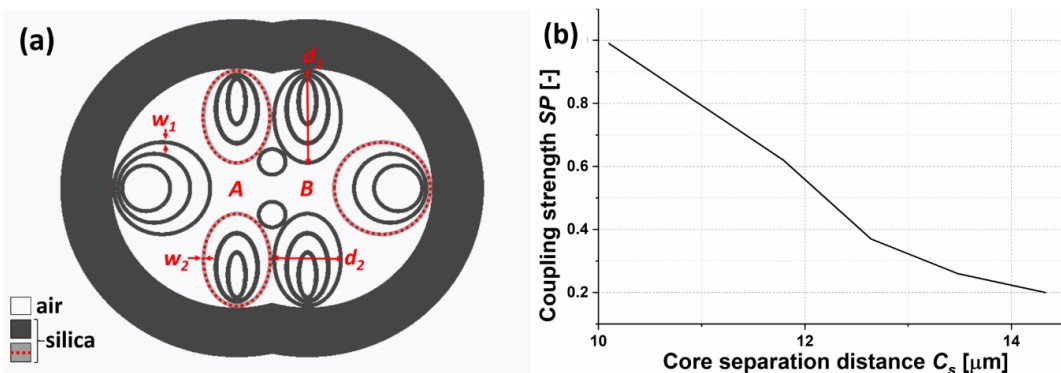

**Figure 5.** (**a**) Cross-section of the fiber with elliptical capillaries. Similarly to Figure 1a–c, the thickened capillaries are grey colored, with red-dotted circles inside. Red letters *A* and *B* indicate the cores, while $w_1$ and $w_2$ are the capillary wall thickness, with the latter being connected to the thinned ones. Due to the introduced ellipticity, two additional dimensions appear—$d_1$ and $d_2$. (**b**) Coupling strength calculated versus the core separation distance. The core separation distance was controlled directly via the dimension $d_2$.

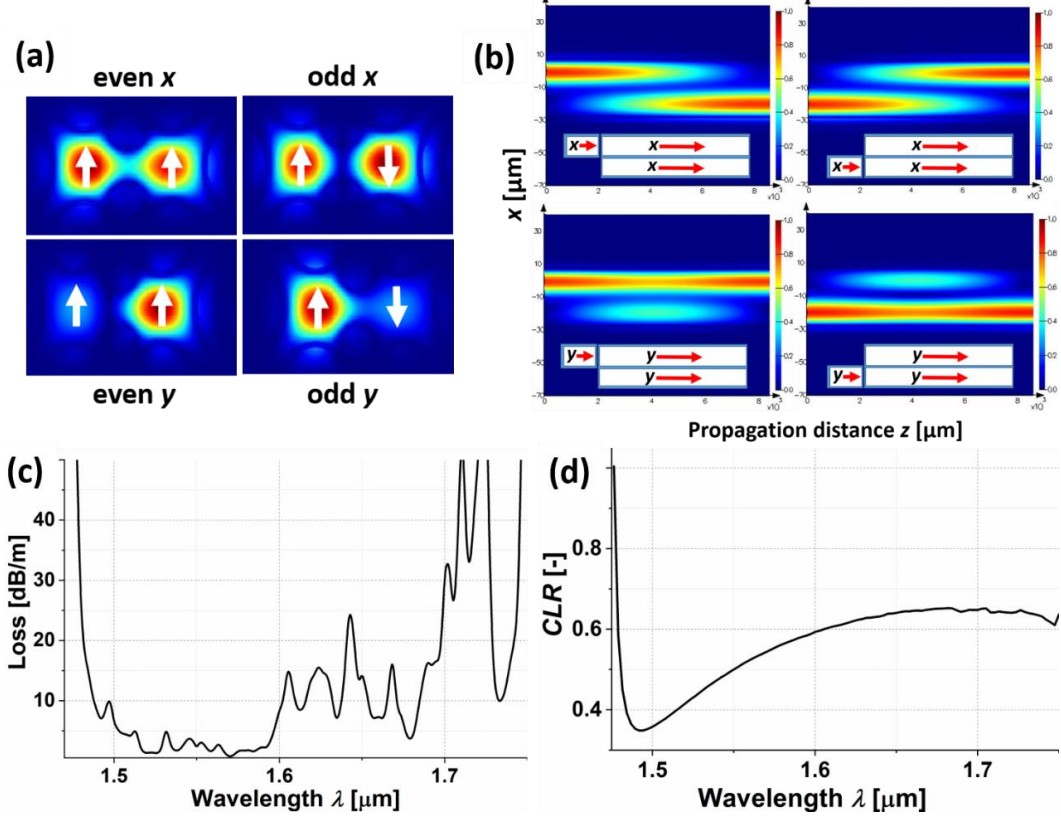

**Figure 6.** (**a**) Distribution of the electric field for the LP01 mode. (**b**) Distribution of the power during propagation. Bottom inserts represent the signal launch setup. (**c**) The highest loss of the fundamental mode LP01 and (**d**) coupling length ratio (*CLR*) versus the wavelength.

## 4. Conclusions

In this work, a new type of a double hollow-core, antiresonant fiber has been presented. The structure was analyzed in terms of using it as a polarization-splitting optical fiber coupler. Using the highly birefringent antiresonant fiber design, a structure with two cores, coupled by omitting a middle microstructure capillary, was created. Important coupling parameters, such as coupling length and coupling ratio, were analyzed. In order to find an optimal solution, three different versions of the DHC-ARF were designed, each having a structural perturbation in the form of thickened walls of the cladding capillaries, either symmetrically in the *x*- or *y*-directions in both cores, or asymmetrically, with one core being perturbed in the *x*-direction and the other one in the *y*-direction. Such modification was justified by finding that for the designed symmetrical DHC-ARFs, the coupling length ratio for the *x*- and *y*-polarized signals could not reach 0.5 or 2, which is necessary for the complete splitting of differently polarized signals. On the other hand, by introducing the asymmetrical modification, this ratio could be obtained within the transmission bands of the DHC-ARF. To further confirm this finding, the analysis of propagation was conducted for the proposed asymmetrical structure. The results have shown that even though the predicted ratios were obtained, the coupling strength was weak for both polarizations, effectively eliminating this structure as a polarization-splitting DHC-HCARF. The reason for that was connected to the weak interaction of the excited supermodes, which were a result of weak mode–field overlap. Since the latter could be effectively increased by reducing the distance between the splitter's cores, the upper and lower DHC-ARF capillaries had their shape changed from circular to elliptical. As a result, the coupling of *x*- polarized odd and even supermodes was greatly increased, while the *y*-polarized signal remained almost the same. Finally, this allowed a DHC-ARF to be prepared with relatively low loss and well-matched coupling lengths for two orthogonal polarizations (8.3 and 4.15 mm for *x*- and *y*-polarized signals, respectively). This resulted in an almost perfect split, with

the coupling strength of the *x*-polarized signal estimated to be ~1 for a core separation distance of 10.1 μm. The extinction ratio of signals with different polarizations after the propagation length of 8.3 mm was calculated to be 16 dB for core A and 17 dB for core B. Perhaps the weakest feature of the proposed structure is its HOMER value, calculated to be 28 at 1550 nm. Further reduction of the high-order mode excitation can be achieved by changing the structure geometrical parameters, such as the diameter of the cladding capillaries or the size of the core. Additionally, one should also note that the structure itself is a challenge from a fabrication point of view, which may hamper its possible fabrication. However, current development of hollow-core fiber technology allows one to safely assume that the appearance of such structures in the future is likely to happen.

**Author Contributions:** Conceptualization, design, and numerical modelling, H.I.S.; data analysis, manuscript preparation, and editing, H.I.S. and M.A.P.

**Funding:** This research was funded by Wroclaw University of Science and Technology grants No. 049U/0032/19 and 0402/0178/18.

**Acknowledgments:** We would like to thank the Wroclaw Centre for Networking and Supercomputing (http://www.wcss.wroc.pl) for letting us conduct the calculations using their computing resources. We would also like to thank our student, Paweł Furmanowski, for his support during this research.

**Conflicts of Interest:** The authors declare no conflict of interest.

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
