# Peer review of "A Dual Hollow Core Antiresonant Optical Fiber Coupler Based on a Highly Birefringent Structure-Numerical Design and Analysis"

_fibers, doi:10.3390/fib7120109_

Round 1

Reviewer 1 Report

The authors considered an interesting problem of creation of a dual hollow core antiresonant fibre (DHC ARF) acting as a polarization fibre coupler. As a result, they demonstrated DHC ARF with rather low loss and well mathced coupling lengths of two orthogonal polarizations. They also demonstrated an almost perfect split with coupling strength of x - polarized signal estimated to be 1 for core separation distance 10 microns. The paper is written in a clear way and contains rather valuable conclusions and results. Nevertheless, I have one question that should be adressed. Can the authors explain why the fiber which was called as a structure - y has minimal loss in Fig.2b? What is the physical reason for this loss behavior? I think the paper can be published in Fibers if the authors answer this question in the paper.

Author Response

Reviewers remarks and questions have been properly addressed and answered in the uploaded .pdf document.

Reviewer 2 Report

The paper is very well written and technical sound.  One recommendation would be to discuss or demonstrate the fabrication of the proposed structure.  

Author Response

Response to Reviewer 2 Comments

Point 1: One recommendation would be to discuss or demonstrate the fabrication of the proposed structure.

Response 1: Indeed, fabrication of such structure would be an ultimate goal in order to verify our proposal. Since it is impossible for us to fabricate such fibre due to the lack of proper fabrication facilities, we were unable to do so, unfortunately. Nevertheless, according to the cited papers and observed development of the hollow-core fibres technology, especially in the field of antiresonant fibres, one can expect such structures to become viable for production in the future. Since this notion is important to determine the structure possible future application, we have included the following sentence in the conclusions section of the manuscript:

Additionally, one should also notice that the structure itself is a challenge from the fabrication point of view, which may hamper its possible fabrication. However, current development of hollow-core fibre technology allows to safely assume that the appearance of such structures in the future is likely to happen.” lines 276-279

In the summary of our Response, we would like to thank the reviewer for the work that has been done during the process of reviewing our paper. Thank you!

Reviewer 3 Report

The paper deals with the design of a new type of dual hollow-core antiresonant fibre (DHC-ARF) employed as a polarization splitter at 1550 nm. Theoretical background was well explained. The device was numerically investigated and its performance evaluated by calculating the coupling length, the coupling length ratio and the extinction ratio. Overall, I think that the paper can be accepted provided that the authors address the following issues:

1) The text does not state if glass refractive index dispersion was taken into account. Was a Sellmeier equation employed?

2) The authors showed how to enhance mode coupling for x-polarized modes. Can the same approach be applied for y-polarized modes too?

Moreover, the following corrections should be performed:

Line 32: "this" should be "that"
Line 33: "one the" should be "one of the"
Line 41: "years the" should be "years, the"
Line 42: "manner" should be "manners"
Line 48: "pressured" should be "pressure"
Line 53: "HCARFs" should be "HC-ARFs"
Line 55: "HCARF" should be "HC-ARF"
Line 71: "single ring" should be "single-ring"
Line 72: "core) the" should be "core), the"
Line 83: "an DHC-ARF" should be "a DHC-ARF"
Line 96: "presented" should be "the presented"
Line 121: "In a general" should be "In general"
Figure 1: the distance between the inner capillaries (0.65R) should be shown in the drawing
Figure 1 caption: "structure_y" is missing
Line 153: "influence" should be "the influence"
Line 157: "As can" should be "As it can"
Line 159: "w1" should be "k1"
Line 162: underlining of "structure_x" is wrong
Line 168: "simulated propagation" should be "simulated the propagation"
Line 180: "this" should be "these"
Line 180: "for the proposed structure its" should be "the proposed structure has a"
Figure 4 caption : "A (a)" should be "A. (a)"
Line 186: "to coupled" should be "to the coupled"
Line 189: "Hence to" should be "Hence, to"
Line 190: "simulation we" should be "simulation, we"
Line 211: "6b distribution" should be "6b, the distribution"
Line 212: the two "lunched" should be "launched"
Line 213: "cores the" should be "cores, the"
Line 213: "shows calculated" should be "shows the calculated"
Line 214: "splitter the" should be "splitter, the"
Line 217: "fibre we" should be "fibre, we"
Line 218: "modes" should be "mode"
Equation 7: is it "HE" or "HOMER"?
Line 220: "This" should be "These"
Line 222: "for the" should be "for a"
Line 222: "mm" should be "mm."
Line 224: "of a" should be "of"
Line 227: "capillary has" should be "capillary, has"
Line 228: "find optimal" should be "find the optimal"
Line 237: "that even" should be "that, even"
Line 239: "DHC-HCARF" should be "DHC-ARF"
Line 244: "loss, well" should be "loss and well"
Line 247: "for core" should be "for a core"
Line 247: "distance 10.1" should be "distance of 10.1"

If possible, please find three equivalent journal papers for references 6, 33 and 42.

Author Response

Response to Reviewer 3 Comments

Point 1: The text does not state if glass refractive index dispersion was taken into account. Was a Sellmeier equation employed?

Response 1: Indeed, the use of a glass refractive index dispersion was not stated. We have used the SiO2(Glass) – Palik model, provided by the Lumerical® materials library. This information has been included in the manuscript in the following sentence:

“To properly calculate the loss and mode profiles of the DHC-ARF, the Palik model (included in the Lumerical® material library) was used to determine the dispersion of SiO2.” lines 160-162

Point 2: The authors showed how to enhance mode coupling for x-polarized modes. Can the same approach be applied for y-polarized modes too?

Response 2: Considering the proposed DHC-ARF structure, the answer is no – purely because of the geometrical reasons (the orientation of the structure and signal’s polarization). As we have presented, the case of best coupling is for the x-polarized modes, which have their polarization parallel to the orientation of the coupling channel between the cores (see line 131). However, what is also worth noticing, the presence of the channel itself was not enough for the asymmetrical structure to provide good coupling strength, as was presented in figure 4. The propagation graphs show that even though the requirement of proper coupling length ratio (i.e. CLR = 2 or 0.5) for both polarizations is fulfilled, the coupling is weak in general. The fields of the x- and y­- polarized odd and even modes are localized only in one of the cores, causing a very small mode field overlap and hence the reduced coupling. To overcome this problem, we have introduced the elliptical capillaries in the y- direction, reducing the distance between both cores and, as one can see in Figure 6a, strongly increasing the ­coupling of the x-polarized modes. In case of the y-polarized modes we also observe an increase in the total coupling, however, it is much smaller than for the x-one. To increase it for the proposed structure, one would have to further reduce the distance between the cores or lift the asymmetry of the cores. However, doing any of the proposed would also affect the x­-polarized signals, hence reducing the structures potential as a polarization beam splitter. Additionally, further reduction of this distance would mean increased ‘flattening’ of the elliptical capillaries, greatly increasing the fabrication difficulties.

Remaining remarks:

All the English corrections and typographical mistakes have been addressed as suggested by the Reviewer.

As for the equivalent journal papers for references [6], [33] and [42] – unfortunately, we were unable to find such. However, in case of [42], the results presented are supported by the previous works of the same authors, similarly to [6]. Reference [33] is the only one without such support, however, as mentioned in the Introduction of the manuscript, it is the only, in our opinion, known report of a dual hollow-core photonic bandgap fibre, which is why we believe it is important and should, as a result, be cited.

In the summary of our Response, we would like to thank the reviewer for the work that has been done during the process of reviewing our paper, both the questions about the paper and the English corrections, which have helped to clarify and increase the overall quality of the paper. Thank you!
